# Use of Thermoregulatory Models to Evaluate Heat Stress in Industrial Environments

**DOI:** 10.3390/ijerph19137950

**Published:** 2022-06-29

**Authors:** Irena I. Yermakova, Adam W. Potter, António M. Raimundo, Xiaojiang Xu, Jason W. Hancock, A. Virgilio M. Oliveira

**Affiliations:** 1International Scientific-Training Centre for Information Technologies and Systems, UNESCO, National Academy of Sciences, 03187 Kyiv, Ukraine; irena.yermakova@irtc.org.ua; 2Thermal and Mountain Medicine Division, U.S. Army Research Institute of Environmental Medicine, 10 General Greene Avenue, Bldg 42, Natick, MA 01760, USA; xiaojiang.xu.civ@mail.mil (X.X.); jason.w.hancock5.civ@mail.mil (J.W.H.); 3Department of Mechanical Engineering, ADAI-LAETA, University of Coimbra, Pólo II da Universidade de Coimbra, 3030-788 Coimbra, Portugal; antonio.raimundo@dem.uc.pt (A.M.R.); avfmo@isec.pt (A.V.M.O.); 4Oak Ridge Institute for Science and Education (ORISE), 1299 Bethel Valley Rd., Oak Ridge, TN 37830, USA; 5Coimbra Polytechnic-ISEC, Rua Pedro Nunes, Quinta da Nora, 3030-199 Coimbra, Portugal

**Keywords:** physiology, biophysics, thermoregulation, heat stress, glass industry

## Abstract

Heat stress in many industrial workplaces imposes significant risk of injury to individuals. As a means of quantifying these risks, a comparison of four rationally developed thermoregulatory models was conducted. The health-risk prediction (HRP) model, the human thermal regulation model (HuTheReg), the SCENARIO model, and the six-cylinder thermoregulatory model (SCTM) each used the same inputs for an individual, clothing, activity rates, and environment based on previously observed conditions within the Portuguese glass industry. An analysis of model correlations was conducted for predicted temperatures (°C) of brain (*T_Brain_*), skin (*T_Skin_*), core body (*T_Core_*), as well as sweat evaporation rate (*ER*; Watts). Close agreement was observed between each model (0.81–0.98). Predicted mean ± SD of active phases of exposure for both moderate (*T_Brain_* 37.8 ± 0.25, *T_Skin_* 36.7 ± 0.49, *T_Core_* 37.8 ± 0.45 °C, and *ER* 207.7 ± 60.4 W) and extreme heat (*T_Brain_* 39.1 ± 0.58, *T_Skin_*, 38.6 ± 0.71, *T_Core_* 38.7 ± 0.65 °C, and *ER* 468.2 ± 80.2 W) were assessed. This analysis quantifies these heat-risk conditions and provides a platform for comparison of methods to more fully predict heat stress during exposures to hot environments.

## 1. Introduction

Working conditions in the glass industry pose significant heat stress on individuals. Mitigation of heat stress in these conditions are important to ensuring a healthy workforce and to avoid risks of injury and heat disorders [1,2,3,4]. Thermal modeling provides a quantifiable and repeatable method of predicting thermal and physiological responses to various conditions and enables the development of data-driven guidance [5,6,7,8]. The present work seeks to quantify the heat-stress conditions of individual workers within the Portuguese glass industry while also comparing the predictions from four thermoregulatory models.

Thermoregulation models are powerful tools that quantitatively represent human responses to a variety of different environmental conditions (e.g., cold to hot, indoors and outdoors). A broad field of potential applications is possible, comprising predictions of the human thermal state over time, prevention of cold/hot hazards, design of mitigation plans, and assessments of physiological adjustments, among many others. Mainly due to computational restrictions, early models were designed to address specific environmental conditions (e.g., just hot or cold). However, these models have become increasingly more sophisticated to provide higher resolution details of human physiological responses, enabling the combination of complex models and the environments they may be applied to [9,10,11,12,13].

Typically, thermoregulation models require several features of input data, embracing at the simplest of forms (1) physical parameters of the environment, (2) human variables, (3) activity levels, and (4) clothing properties. Critical inputs of the environmental conditions include air temperature (*T_a_*), relative humidity (*RH*), wind velocity (*V_a_*), and the surrounding area temperatures (represented by radiant or mean radiant temperature (*T_r_*, *T_mr_*)) and specific forms of impinging radiation (solar, etc.). The basics of the human parameters include two parts: their features (e.g., sex, stature (height), body mass, hydration status, food intake, and acclimatization status) and their activity (e.g., posture (e.g., standing, sitting) and metabolic rate (resting or active)). Clothing properties (e.g., dry and evaporative resistance, weight, textile type) are a critical element, as they represent an additional resistance layer for heat and water vapor exchange between the human body and the environment.

Validation of thermoregulation models is an important step to provide confidence in any use and interpretation of results and as a platform for making continued scientific improvements to these types of methods [14,15,16,17]. Thorough attention is usually given to this phase, while at the same time, equal importance should be given to the characterization of the environment and the measurement or estimation of the person features, physical activity, and clothing properties. If any of these approaches are neglected, the “quality” of the numerical predictions might be compromised. Conducting comparisons represents one more stage towards improvement of methods and the expansion to models capable of predicting a wide range of conditions [18,19,20,21,22].

This study compared the simulated outputs of four well-established, rationally designed thermoregulatory models: the health-risk prediction (HRP) model [23], the human thermal regulation model (HuTheReg) [18,24], the SCENARIO model [25,26,27], and the six-cylinder thermoregulatory model (SCTM) [28,29]. The use of multiple models is helpful for highlighting the range of likely outcomes while also providing quantified values to the level of heat stress and likely risk of individual workers within these conditions. While there are heat-stress guidance and standards for general workplaces and activities [30,31,32,33], extensive reviews of occupational settings [34,35,36,37,38,39,40], and multiple indices [21,22,41,42,43,44], this work seeks to address the specific need for quantitative guidance for these unique conditions of the glass industry.

## 2. Methods

### 2.1. Design

Each of the four thermoregulatory models (HRP, HuTheReg, SCENARIO, and SCTM) were used to simulate responses to observed heat-stress conditions within the Portuguese glass industry. Comparisons were made on the predicted brain, skin, and core body temperatures (*T_Brain_*, *T_Skin_*, *T_Core_*) as well as the rate of heat loss from sweat evaporation (*ER*). These physiological variables are needed to adequately analyze heat effects on humans. Comparisons were performed between time series predictions for each measure as well as for mean ± SD values from each phase of activity. Mean ± SD are shown and compared between the models, and the Pearson correlation coefficients were also calculated between the model predictions during exposures.

#### 2.1.1. Framework

The four models were used to simulate an entire workday of 8 h (480 min) in the glass industry, for which all models used the same inputs of person features, physical activity, clothing properties, and environment characteristics. These data were taken from the field survey carried out by Oliveira et al. [45], where 19 workplaces pertaining to five Portuguese glass facilities were checked. Oliveira et al. [45] assessed the heat stress using the wet bulb globe temperature (WBGT) index and the predicted heat strain (PHS) model [30,31,46,47]. These workplaces were characterized from a human thermal perspective as favorable, acceptable, critical, and very critical. However, to avoid excessive risk of heat stress, the workers only stayed in critical and very critical workplaces for limited periods of time. Therefore, during the entire workday, each worker performs a specific “working profile” involving different workplaces.

#### 2.1.2. Working Profiles

For practical purposes, only two glass industry working profiles were selected, more precisely, one involving workplace G12, classified by Oliveira et al. [45] as acceptable, and other involving workplace G14 (which was rated as very critical). In this study, to allow the readers an easy reference to the work of Oliveira et al. [45], the working profile involving workplace G12 is called as G_12, and the one involving workplace G14 is called as G_14. The sex of the worker was assumed as male (the most common scenario in the Portuguese glass industry), and working profiles comprising nine phases were considered (Table 1): arrival, work 1, work 2, work 1, mealtime, work 1, work 2, work 1, and departure. Workplace G_12 includes moderate heat-stress conditions, while workplace G_14 includes extreme heat-stress conditions (Table 2). The difference between the two working profiles considered occurs during the work 1 phase, where the worker is required to conduct work in moderate or extreme heat-stress conditions. Global characteristics of the surrounding environment of each workplace are shown in Table 2 for both working profiles. The overall properties of clothing worn by workers in each exposure period are shown in Table 3.

### 2.2. Simulation Inputs 

#### 2.2.1. Human and Clothing Inputs

Human and clothing inputs were standardized between each of the models (Table 1, Table 2 and Table 3). The simulated worker was assumed as 1.69 m in height, a body mass of 74 kg, with an associated body surface area of 1.84 m^2^, a resting heart rate of 65 bpm, and resting systolic pressure of 120 mm.c.Hg. Resting and active metabolic rates (*M*) were estimated following the methods of level II of accuracy of ISO 8996 [32] by adding the metabolic rates corresponding to the posture, the type of work, the body motion related to the work speed and the basal metabolic rate for each single activity. However, it is important to note that internally each of the models’ approach to making these calculations differed slightly.

Clothing inputs varied in units and format for each model; however, each of them used the same fundamental values as inputs. Based on ISO 9920 [33] and on the work from Oliveira et al. [45], standardized clothing inputs of thermal insulation (clo, m^2^·°C/W), evaporative resistance (m^2^·kPa/W), radiative emissivity, mass (kg), and body surface coverage were used (Table 3). It is important note that “global values” correspond to the weighted total of the clothing on the human. That is to say, clothing parameters (e.g., thermal insulation, evaporative resistance, radiative emissivity, and mass) are functionally different values for the different body parts based on clothing coverage in that area.

#### 2.2.2. Environmental Conditions

Oliveira et al. [45] measured the physical parameters of the environment according to ISO 7726 [48] using equipment from Brüel and Kjær and from Testo. In the case of the former, the WBGT-Heat Stress Monitor type 1219 was used to measure the natural wet bulb (*T_nw_*), the 150 mm globe (*T_g_*), and the air (*T_a_*) temperatures. The air and globe temperatures were then used to estimate the mean radiant temperature (*T_mr_*) according to the expression suggested in ISO 7726 [48,49]. For the air velocity (*V_a_*), despite being measured (hot sphere sensor from Testo (ref. 0635 1049) connected to the data logger Testo 445 (ref. 0560 4450)), a value of 0.5 m/s was considered as representative of the type of workplaces under analysis. A humidity transducer also from Testo (ref. 0636 9741) was also connected to Testo 445. All measurements of the physical parameters were preceded by a stabilization period of 30 min.

#### 2.2.3. Activity Phases and Exposure Times

Activity levels and phases of exposure are categorized by three periods of heat-stress potential (favorable, moderate, and stressful) and are simulated collectively over the nine phases that represent a 480 min (8 h) work day. These three types represent different environmental and activity-based characteristics, where the favorable periods represent low-intensity, low-exposure phases, shown functionally as arrival, mealtime, and departure (phases 1, 5, and 9) (Table 1). Moderate periods involve working activities in an acceptable environment and are called work 2 (phases 3 and 7). Stressful periods are referenced as work 1 (phases 2, 4, 6, and 8) and involve a high level of physical activity in a hot (G_12) or extremely hot (G_14) environment.

The exposure period of each phase is listed in Table 1. To more easily follow the path of the worker throughout the different sites that comprise the working day, greyscale patterns in Table 1 are used in the results figures for each modeled simulation. For modeling purposes, the transition between environmental conditions occurs suddenly.

### 2.3. Rational Models Assessed

#### 2.3.1. Health Risk Prediction (HRP) model

The health-risk prediction (HRP) model has a traditional structure of the human thermoregulatory system and is derived mainly from mechanistic methods, as the main predictions are calculated based on a series of equations built on a rational construct [29]. This model consists of an active part, including a regulatory center with thermal responses, and passive processes related to heat production, which are distributed through the body by conduction and convective transfer by blood and then exchanged with the environment by radiation, convection, and sweat evaporation. The model considers the human body divided into 14 parts (13 cylinders and 1 sphere) and 39 compartments (38 layers plus a blood compartment) taking into account right and left extremities.

Rational calculations of the HRP model are fairly extensive and are designed to output regional and total body temperature responses to include local skin (14 locations) and mean skin temperature, brain, blood, internal organs, muscles, and fat temperatures (°C). Cardiovascular outputs from the HRP model include stroke volume (*SV*, mL), cardiac output (*CO*, L/h), and heart rate (*HR*, bpm). The model also allows to simulate the effect of different clothing, as it contains large database of various fabrics for composing of clothing and protective garment [50].

The HRP model has been used for modeling of human immersion in cold water [51] and in warm water [52] and validated for hot, humid conditions [20].

#### 2.3.2. Human Thermal Regulation (HuTheReg) Model

The human thermal regulation (HuTheReg) model is a rationally designed to allow for simulations of human thermophysiological responses to a wide range of environmental conditions [8,24]. This program was implemented considering only the male sex and is composed by several modules, namely (i) male thermophysiological response, (ii) heat and water vapor transport through the clothing, (iii) heat and water exchange between the external surface of clothing (or skin) and the environment and surroundings, (iv) start and evolution of skin injuries (pain and burn), and (v) detection of specific drawbacks within the human being. Due to its interdependency, all modules run iteratively in each time step until a specific convergence criterion is reached. The main module of HuTheReg is based on work from Stolwijk [9], while more recent work has been used to expand to include additional modules for improved capabilities [11,53]. This model considers the human body divided in 22 segments (face, scalp, neck, chest, abdomen, upper back, lower back, pelvis, left shoulder, right shoulder, left arm, right arm, left forearm, right forearm, left hand, right hand, left thigh, right thigh, left leg, right leg, left foot, and right foot). Each body part is comprised of four layers (core, muscle, fat, and skin), collectively a total of 88 nodes, plus an additional node (89th) corresponding to the central blood compartment. The passive thermophysiological phenomena and the active thermoregulatory responses (shivering and vasoconstriction or sweating and vasodilatation) are simulated for each specific human body segment but considering its influence and interdependence with the global thermal state of the body. Each run can simulate up to 60 consecutive scenarios (phases), each one representing different conditions in terms of posture, orientation, activity, intake of food/drinks, clothing, and thermo-hygrometric environment characteristics. Some inputs of each phase are specified for whole human body and others for each body segment (each one considered completely nude or completed dressed).

As it is embodied in software form, the HuTheReg allows for a significant number of output calculations, both for whole body, regional elements, and for physiological calculations. The validation of the HuTheReg software was performed comparing the program’s predictions with experimental results. The validation process spanned a wide-range of conditions, which included different kinds of thermo-hygrometric environments, exposures, exercise intensities, and clothing [8,18]. A good agreement was achieved, which indicates an interesting capacity of the program to predict the thermophysiological response of the human body to a wide variety of conditions.

#### 2.3.3. SCENARIO Thermal Model

SCENARIO thermal model is a single-cylinder, rationally based model that consists of seven compartments and is made up of five concentric cylinders that represent human core, muscle, fat, and vascular and avascular skin plus a central blood compartment and a clothing layer [25,26,27]. The SCENARIO model combines physiologically based variables and biophysical calculations to make time-series predictions for a given human, set of activities, and environmental exposures (i.e., scenarios). SCENARIO was developed by Kraning and Gonzalez [25,26,27,54] and has been recently enhanced by Tan et al. [55]. However, the model has some basis foundation from a number of key sources, namely Wyndham and Atkins [56,57], Gordon et al. [58], Stolwijk and Hardy [59], Stolwijk [9,60], Montgomery [61], Montgomery and Williams [62], Werner et al. [63,64,65], Gagge et al. [66,67,68], and Wissler [69,70]. The model requires several inputs for individual characteristics (e.g., anthropometrics, health status), environmental conditions, clothing properties (biophysics), and activity types to generate physiological predictions (metabolism, heart rate, cardiac output, stroke volume, skin, and core body temperature) over a given time course.

#### 2.3.4. Six Cylinder Thermoregulatory Model (SCTM)

The six-cylinder thermoregulatory model (SCTM) is a rational model based on the first principles of physiology and the physical laws of heat transfer [28,29]. SCTM considers the human body subdivided into six segments representing the head, trunk, arms, legs, hands, and feet. Each segment is further divided into concentric compartments representing the core, muscle, fat, and skin. The integrated thermal signal to the thermoregulatory controller is composed of the weighted thermal input from thermal receptors at various sites distributed throughout the body. The difference between this signal and its threshold activates the thermoregulatory actions: shivering heat production, vasodilation/vasoconstriction, and sweat production. The SCTM has been validated for a broad range of conditions, including heat, cold, and water immersion [71,72]. SCTM has been used to evaluate heat strain in personal protective equipment [73] and design personal cooling systems [74]. It has been used to develop user friendly tools for operational use, probability of survival decision aid (PSDA) [75], and cold ensemble decision aid (CoWEDA) [15]. SCTM inputs include individual characteristics, intensity of activity, environmental conditions, and clothing properties (i.e., thermal resistance and evaporative resistance) for each of the six body regions. SCTM predicts physiological responses (e.g., core temperatures, skin temperatures, and sweat rates for six body regions).

### 2.4. Statistical Analysis

Statistical analyses were performed using MATLAB (The MathWorks, Inc., Natick, MA, USA). Descriptive statistics are presented as means ± SD. Pearson correlation coefficients were calculated between the model predictions during exposures.

## 3. Results

Figure 1 shows the comparison of each models’ prediction of brain (*T_Brain_*), skin (*T_Skin_*), and core (*T_Core_*) temperatures and sweat evaporation rate (*ER*) over the entire work shift for the moderate working profile (G_12), while Figure 2 shows these same predictions for the stressful working profile (G_14). Additionally, Table 4 and Table 5 show the mean ± SD values for phases 2, 4, 6, and 8 for G_12 (Table 4) and G_14 (Table 5) working profiles.

Exposure to both the favorable (arrival, mealtime, departure) and work 2 (moderate working periods) conditions represent the lower heat stress, while work 1 (stressful working periods) corresponds to the most significant exposure in each working condition (G_12 and G_14). As the work 1 exposures have the highest associated risk and happen four times during the day, the analyses focus mostly on these conditions. The onset of responses to exposure to work 1 conditions can be clearly seen as a sharp increase in every parameter, followed by a decrease as the worker moves into other thermal environment (favorable or work 2 conditions). Despite the differences shown by the thermoregulation models (e.g., maximum values), the pattern in all four models is consistently the same; moreover, the mean values between each exposure are surprisingly close.

Focusing the analysis on the maximum values, it is interesting to see that despite the model, the maximum values are reached at the end of each exposure to work 1 conditions (Figure 1 and Figure 2). Additionally, the absolute maximum is typically reached at the end of the day during the fourth exposure to work 1. This was true for the absolute predicted maximum values (*T_Brain_*, *T_Skin_*, *T_Core_*, and *ER*) of each model during the moderate working profile (G_12): HRP (*T_Brain_* 38.12 °C, *T_Skin_* 37.67 °C, *T_Core_* 38.08 °C, *ER* 324.32 W), HuTheReg (*T_Brain_* 38.93 °C, *T_Skin_* 37.76 °C, *T_Core_* 38.9 °C, *ER* 263.55 W), SCENARIO (*T_Skin_* 36.41 °C, *T_Core_* 37.57 °C, *ER* 169.73 W), and SCTM (*T_Brain_* 38.45 °C, *T_Skin_* 37.30 °C, *T_Core_* 38.49 °C, *ER* 303.97 W). This trend also held true for the absolute predicted maximum values (*T_Brain_*, *T_Skin_*, *T_Core_*, and *ER*) of each model during the stressful working profile (G_14) for each model: HRP (*T_Brain_* 39.08 °C, *T_Skin_* 39.39 °C, *T_Core_* 39.02 °C, *ER* 575.23 W), HuTheReg (*T_Brain_* 40.52 °C, *T_Skin_* 40.48 °C, *T_Core_* 40.70 °C, *ER* 687.61 W), SCENARIO (*T_Skin_* 38.59 °C, *T_Core_* 39.30 °C, *ER* 561.80 W), and SCTM (*T_Brain_* 41.53 °C, *T_Skin_* 41.25 °C, *T_Core_* 41.38 °C, *ER* 491.76 W).

Comparisons of mean values show that for each model, the four phases of exposure to work 1 are similar (Table 4 and Table 5). Considering the moderate working profile (G_12) and *T_Core_*, the maximum mean value is 37.51 °C (HuTheReg), and the minimum mean value is 37.28 °C (SCENARIO) in phase 2; for the remaining phases, the maximum and minimum mean values are 38.26 °C (HuTheReg) and 37.48 °C (SCENARIO) (phase 4); 37.94 °C (HuTheReg) and 37.41 °C (SCENARIO and HRP) (phase 6); and 38.33 °C (HuTheReg) and 37.48 °C (SCENARIO) (phase 8) (Table 4). For the stressful working profile (G_14), the maximum and minimum mean values are obviously higher between the *T_Core_* maximum mean and the minimum mean values: 38.69 °C (HuTheReg) and 37.86 °C (HRP) (phase 2); 39.64 °C (HuTheReg) and 38.22 °C (HRP) (phase 4); 39.40 °C (HuTheReg) and 37.96 °C (HRP) (phase 6); and 39.73 °C (HuTheReg) and 38.23 °C (HRP) (phase 8) (Table 5).

Due to the extreme conditions of work 1 phases of working profile G_14, it is also important to look at the values of the sweat evaporation rate (*ER*); Table 5 highlights that the minimum mean value predicted by the models is 311.16 W (SCTM, phase 2), and the maximum mean value is 590.36 W (HuTheReg, phase 8). These values correspond to a significant loss of liquids (≅494 and 1092 g/h, respectively), thus issuing the need to pay particular attention to liquids intake throughout the day, a topic that, despite its true requirement in hot environments, is most often neglected. 

Table 6 and Table 7 show the calculated Pearson correlation coefficients between each of the models for both the moderate (G_12) (Table 6) and stressful working profile (G_14) (Table 7). Each of the models had a highly correlated between each other (0.8–0.9), while others were very highly correlated (0.9–1.0). The highest correlations for each working profile showed similar patterns between models, where in both G_12 and G_14, *T_Brain_* was most correlated between SCTM and HuTheReg and *T_Skin_* between SCTM and HRP. For G_12, the highest correlation for *T_Core_* was between SCENARIO and HuTheReg (Table 6), while for G_14, the highest correlation for *T_Core_* was between SCENARIO and HRP (Table 7). Correlations for *ER* were high in moderate conditions (G_12) between SCENARIO and HRP and between SCTM and HuTheReg (Table 6), while for the stressful working profile (G_14), they were highest between SCENARIO and HuTheReg (Table 7).

## 4. Discussion

The present work modeled human responses to significant heat-stress exposure, which is a regular practice for the glass and ceramics industries. The extreme conditions for these workers undoubtedly impose a significant heat-stress burden; this work used four previously validated models to quantify the level of physiological strain imposed on the individuals. For the two working profiles considered (moderate G_12 and stressful G_14), clinical temperature thresholds were observed to indicate risks of heat exhaustion (best case) and of heat stroke (worst case). Using *T_Core_* as a marker, in the moderate profile (G_12), most of the models (all except SCENARIO) predicted maximal temperatures indicative of potential heat exhaustion (>38 °C), while in the stressful profile (G_14), all models predicted maximal temperatures indicative of potential heat stroke (>39 °C) [76].

This study highlights some of the critically valuable information that can be gained from using thermoregulatory modeling for mitigating and planning for heat-stress conditions. These analyses showed that the working conditions on the glass industry can lead the thermal status of the human body to a high hyperthermic status, which has the potential to the occurrence of harmful incidents with the workers.

There are several limitations to the current work. The use of thermoregulation models should represent a contribution to produce useful guidelines. While the mean ± SD values could be easily calculated for the entire working day (encompassing all exposure), this analysis introduces some bias and leads to mischaracterization of the exposure. Therefore, the most critical exposures during the work shift were considered, allowing for more accurate assessment of the burden/strain and its evolution in time imposed by the thermal environment on the worker throughout the day. Additionally, given their complexity, the full details of each model were not presented, as describing them all would take away from the main goal of this work, and the number of varied and unique characteristics would make this less-than helpful for use for health and safety applications.

While some key differences between each of the models have been highlighted within the manuscript, it is important to note several others. Each of the models’ input and output variables differ slightly. Some of these differences in how the models interpret responses can be seen by looking at Table 4 and Table 5 in combination with Figure 1 and Figure 2. One example can be seen in Table 4 and Figure 1, where HRP has lower deviation of temperature values but higher rates of evaporative rates compared to the other models. While it is unclear in the current work if this improves the accuracy of the model compared to the others, it shows how this higher rate of evaporation allows for more stable predicted temperatures. Additionally, the inputs for clothing properties differ between the models, as the HRP, HuTheReg, and SCTM all use regional values of clothing biophysical inputs (representing their respective model nodes), while SCENARIO uses a global (total) value to represent a weighted measure for the total body. Moreover, the algorithms used for the simulation of heat and moisture transport through clothing and between the clothing (or skin for nude elements) and the environment differs from model to model. Inputs for the environmental conditions differ slightly also, where SCENARIO, SCTM, and HuTheReg use both air temperature (*T_a_*) and mean radiant temperature (*T_mr_*) as inputs, while HRP used an operative temperature, which the mean between the *T_a_* and *T_mr_*. Additionally, the output value of *T_Core_* differs in physical definition between the models, where HuTheReg considers this temperature of the intestine, and HRP, SCENARIO, and SCTM consider a rectal temperature. Similarly, *T_Brain_* in HuTheReg corresponds to hypothalamus temperature and to actual brain temperature in HRP and SCTM, while SCENARIO does not predict it.

In almost all heat-stress conditions, an important part of heat imposed on the human body comes from metabolism. While each model functionally accounts for activities and makes predictions of metabolic costs in different ways, for the present work, all models used the same inputs to represent those estimated from prior work. However, it is important to note that internally each of the models’ approach to making these calculations differed slightly. As metabolic heat production represents the largest influence on heat stress, it is important to accurately make these predictions (e.g., if these are incorrect, the model will be systemically impacted). For the current work, inputs for the metabolic heat production (i.e., activity rates) were performed based on methods outlined in ISO 8996 [32]. However, making more accurate and individualized methods could be used to aid in these predictions.

There are many reasons that justify differences between the models based on their origins, intended use cases, data used for development, and mathematical structures. These differences were expected along with some obvious differences in predictions. However, despite the intrinsic characteristics of each model, once validated, these models do represent powerful tools to mitigate and even to avoid heat disorders and hazards during exposures to very different thermal environments comprising both cold and hot exposures and a range of occupational (e.g., industrial, firefighting, sports) and military contexts. The present work provides one step closer to a multi-model approach to assessing these complex and different environments.

Although the comparison between the four thermoregulatory models was based only on temperatures of the brain, skin, core body, and on sweat evaporation rates, each of these models produce a wide range of other additional thermophysiological parameters. These added parameters can also be considered during the development of safety plans for working in hot environments and on ensuring prevention of workers from heat injuries and life-threatening exposures. For example, software-embodied versions of HRP, HuTheReg, and SCTM allow for output values for both the human body as a whole and for each of the body segments (i.e., core, muscle, fat, skin, and clothing temperatures) as well as physiological outputs as a whole or regionally (i.e., metabolism, heat stored and flux-rates of heat, of sweat, of water, and of work). Additionally, these models are able to provide quantitative interpretations of things such as thermal comfort or even health or injury implications (i.e., detection of heat-related disorders within the person (introversion, heat stroke, permanent brain damage, death) and skin pain, skin burn areas, and corresponding degree).

It is important for continued studies tailored to specific environments, activities, and geolocations [37,77,78,79,80,81,82,83,84] to ensure appropriate guidance can be developed to protect individuals. While it is important to note that models, guidance, and decisions aids provide significant values, ideally, direct measures from individuals should be a goal to allow for more accuracy and real-time heat-stress risk mitigation [85,86,87,88].

This work represents one outcome of a cooperation that involved researchers from very different countries and cultures. During this process, the authors worked together and shared knowledge towards a single purpose: the development of common efforts to mitigate injuries and even casualties during exposures to severe thermal environments and to improve health and safety of working conditions. In the current days, this is a statement that, in itself, despite not adding any scientific value to the present paper, the authors would like to emphasize.

## 5. Conclusions

This analysis shows that the working conditions of the glass industry pose significant risks of hyperthermia or, at least, have the potential to impose unsafe or harmful conditions for workers. Specifically, results quantify that the severe working conditions considered in this study lead to a very significant fluid loss (~1000 g/h), highlighting the need to pay particular attention to fluid intake throughout the working day. Moreover, if *T_Core_* is used as a heat-stress marker, three of the four models predicted maximal temperatures higher than 38 °C in the moderate environment, revealing a potential heat exhaustion condition, while in the more extreme environment, all models predicted maximal temperatures higher than 39 °C, representing a serious risk of heat stroke. Pearson correlation coefficients between each of the models were highly (0.8–0.9) and very highly (0.9–1.0) related. Despite these encouraging results, the limitations of this work, being mainly related with the use of mathematical models to reproduce and simulate human behaviors, are emphasized. Finally, it is also important to stress that the full details of each model are not presented and discussed, as describing them all in detail would turn away the attention of the readers from the main goal of this work. The authors hope that the present manuscript might be helpful for use for health and safety applications, namely in hot thermal environments.

## Figures and Tables

**Figure 1 ijerph-19-07950-f001:**
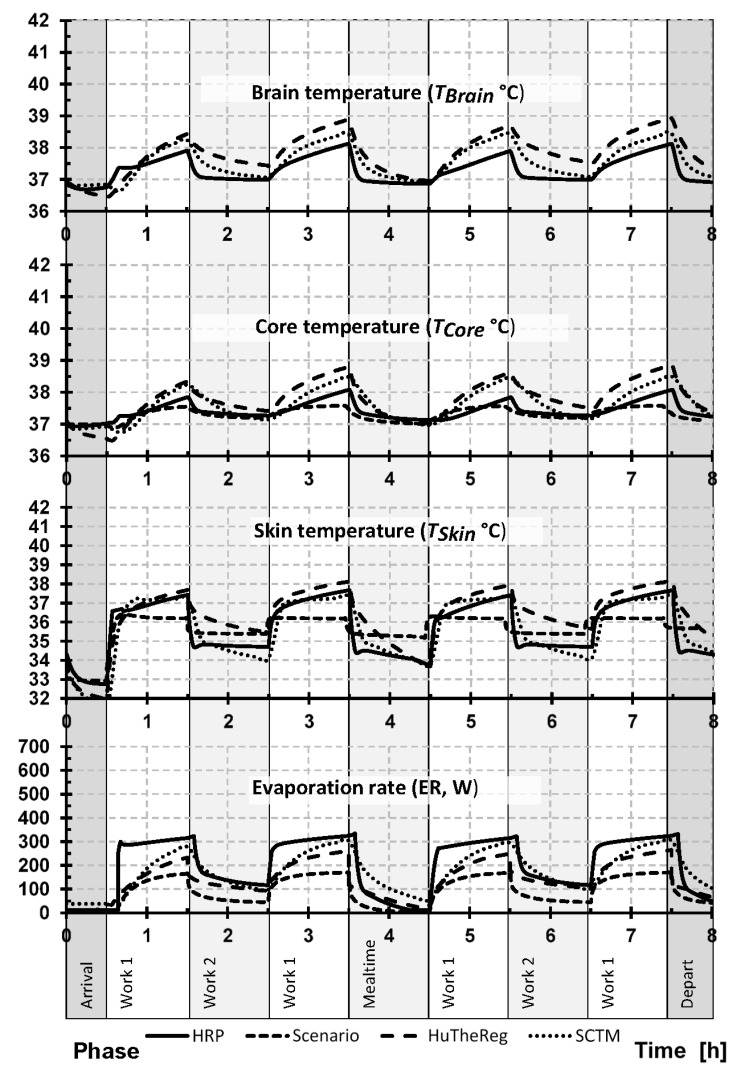
Predicted response pattern of brain, skin, and core body temperatures and evaporation rate during the entire work shift for moderately hot conditions (G_12). Note: HRP, health-risk prediction (HRP) model; HuTheReg, human thermal regulation model (HuTheReg); SCTM, six-cylinder thermoregulatory model.

**Figure 2 ijerph-19-07950-f002:**
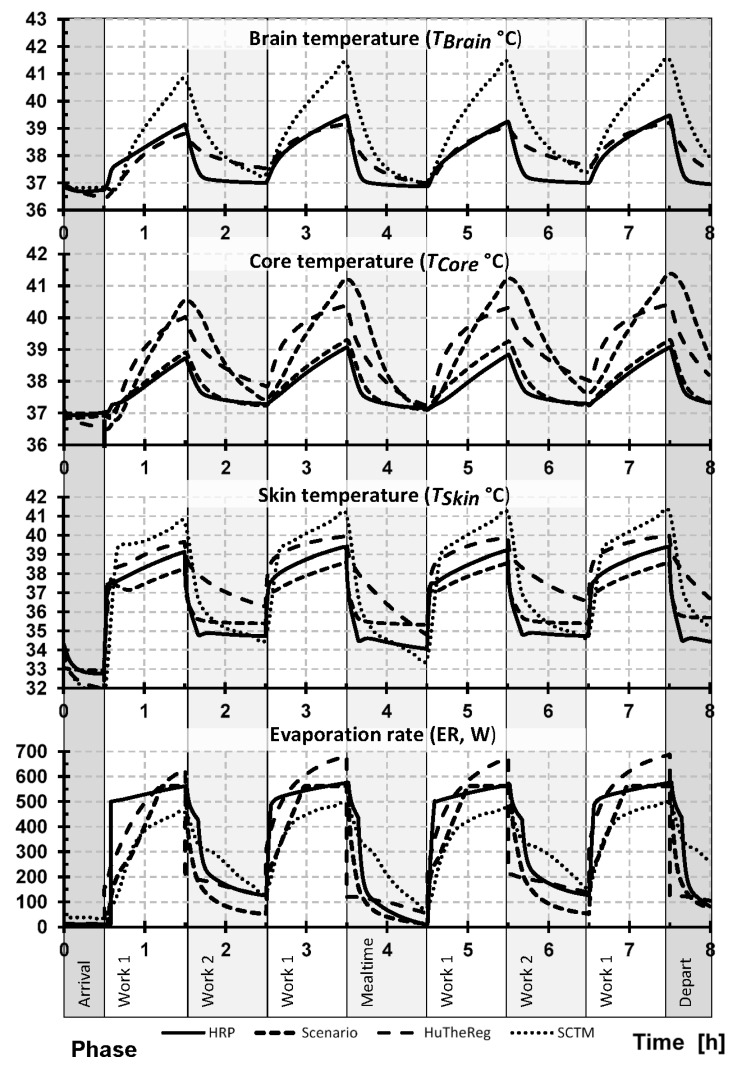
Predicted response pattern of brain, skin, and core body temperatures and evaporation rate during the entire work shift for extremely hot conditions (G_14). Note: HRP, health-risk prediction (HRP) model; HuTheReg, human thermal regulation model (HuTheReg); SCTM, six-cylinder thermoregulatory model.

**Table 1 ijerph-19-07950-t001:** Characteristics of the nine phases of the working day for working profiles G_12 and G_14.

Phase *n*°	1	2	3	4	5	6	7	8	9
Phase Name	Arrival	Work 1	Work 2	Work 1	Mealtime	Work 1	Work 2	Work 1	Departure
Exposure (min)	30	60	60	60	60	60	60	60	30
Metabolic rate (met)	1.20	2.23	2.00	2.23	1.00	2.23	2.00	2.23	1.20
Body posture	Standing	Standing	Standing	Standing	Sitting	Standing	Standing	Standing	Standing
Mass of liquids intake (kg)	0.00	0.00	0.25	0.25	0.50	0.25	0.25	0.25	0.25
Temperature of the liquids intake (°C)	0.00	0.00	20.00	20.00	20.00	20.00	20.00	20.00	20.00
Liquid specific heat (J/(kg·°C))	0.00	0.00	4.186	4.186	4.186	4.186	4.186	4.186	4.186

**Table 2 ijerph-19-07950-t002:** Global characteristics of the surrounding environment for working profiles G_12 and G_14.

Moderate (G_12) and Extreme Heat Stress (G_14) Working Profiles							
Favorable Period (Phases 01 (Arrival), 05 (Mealtime) and 09 (Departure))							
***T_a_*** (°C)		***RH*** (%)		***V_a_*** (m/s)		***T_mr_*** (°C)	
20.00		60.00		0.50		20.00	
**Moderate period (phases 03 and 07 (work 2))**							
25.00		50.00		0.50		25.00	
**Moderate heat stress working profile (G_12)**				**Extreme heat stress working profile (G_14)**			
**Stressful periods: phases 02, 04, 06, and 08 (work 1)**				**Stressful periods: phases 02, 04, 06, and 08 (work 1)**			
***T_a_*** (°C)	***RH*** (%)	***V_a_*** (m/s)	***T_mr_*** (°C)	***T_a_*** (°C)	***RH*** (%)	***V_a_*** (m/s)	***T_mr_*** (°C)
32.10	30.30	0.50	71.80	61.10	6.80	0.50	102.00

**Table 3 ijerph-19-07950-t003:** Global clothing properties for working profiles G_12 and G_14. (Favorable (phases 01, 05, and 09), moderate (phases 03 and 07), and stressful periods (phases 02, 04, 06, and 08)).

Basic Thermal Insulation	Evaporative Resistance	Vapor Permeability Efficiency	Radiative Emissivity	Mass	Specific Heat
(clo; m^2^·°C/W)	(m^2^·kPa/W)	(N.D.)	(N.D.)	(kg)	(kJ/kg·°C)
0.603 0.093	0.700	0.478	0.906	1.3	1.0

**Table 4 ijerph-19-07950-t004:** Mean and standard deviation values for moderate working profile G_12.

		*T_Brain_* (°C)	*T_Skin_* (°C)	*T_Core_* (°C)	*ER* (W)
**Phase 2**	**HRP**	37.49 ± 0.27	36.80 ± 0.68	37.43 ± 0.23	257.42 ± 101.04
	**SCENARIO**	-	36.05 ± 0.65	37.28 ± 0.23	111.22 ± 56.21
	**HuTheReg**	37.58 ± 0.62	36.69 ± 1.15	37.51 ± 0.61	151.37 ± 61.98
	**SCTM**	37.49 ± 0.60	36.21 ± 1.98	37.44 ± 0.55	164.52 ± 92.14
**Phase 4**	**HRP**	37.69 ± 0.30	37.11 ± 0.50	37.67 ± 0.25	300.55 ± 29.98
	**SCENARIO**	-	36.18 ± 0.11	37.48 ± 0.11	153.01 ± 23.03
	**HuTheReg**	38.33 ± 0.43	37.57 ± 0.51	38.26 ± 0.41	218.35 ± 38.05
	**SCTM**	37.92 ± 0.46	36.71 ± 1.08	37.88 ± 0.47	233.73 ± 69.69
**Phase 6**	**HRP**	37.45 ± 0.27	36.74 ± 0.57	37.41 ± 0.24	280.75 ± 51.42
	**SCENARIO**	-	36.20 ± 0.15	37.41 ± 0.17	139.06 ± 35.88
	**HuTheReg**	38.00 ± 0.54	37.18 ± 0.83	37.94 ± 0.51	188.55 ± 54.26
	**SCTM**	37.82 ± 0.52	36.64 ± 1.17	37.77 ± 0.53	210.71 ± 81.27
**Phase 8**	**HRP**	37.69 ± 0.30	37.11 ± 0.50	37.67 ± 0.25	300.53 ± 29.99
	**SCENARIO**	-	36.18 ± 0.11	37.48 ± 0.11	153.03 ± 23.01
	**HuTheReg**	38.40 ± 0.41	37.64 ± 0.48	38.33 ± 0.40	224.05 ± 35.79
	**SCTM**	37.94 ± 0.45	36.72 ± 1.06	37.91 ± 0.46	236.91 ± 68.30

**Table 5 ijerph-19-07950-t005:** Mean and standard deviation values for stressful working profile G_14.

		T_Brain_ (°C)	T_Skin_ (°C)	T_Core_ (°C)	ER (W)
**Phase 2**	**HRP**	38.24 ± 0.60	38.18 ± 0.87	37.86 ± 0.50	490.83 ± 140.63
	**SCENARIO**	-	37.46 ± 0.79	37.93 ± 0.61	378.26 ± 173.47
	**HuTheReg**	38.76 ± 1.15	38.65 ± 1.30	38.69 ± 1.14	460.59 ± 136.33
	**SCTM**	38.80 ± 1.47	38.71 ± 3.00	38.53 ± 1.36	311.16 ± 151.78
**Phase 4**	**HRP**	38.58 ± 0.65	38.57 ± 0.72	38.22 ± 0.53	530.14 ± 65.97
	**SCENARIO**	-	37.80 ± 0.58	38.39 ± 0.60	469.68 ± 128.62
	**HuTheReg**	39.71 ± 0.73	39.37 ± 0.63	39.64 ± 0.70	577.87 ± 97.77
	**SCTM**	39.62 ± 1.35	39.39 ± 2.08	39.38 ± 1.29	395.25 ± 117.27
**Phase 6**	**HRP**	38.33 ± 0.62	38.31 ± 0.75	37.96 ± 0.52	513.78 ± 85.69
	**SCENARIO**	-	37.76 ± 0.58	38.33 ± 0.61	460.42 ± 136.00
	**HuTheReg**	39.54 ± 0.92	39.18 ± 0.85	39.40 ± 0.87	545.15 ± 118.28
	**SCTM**	39.52 ± 1.48	39.19 ± 2.44	39.22 ± 1.42	361.09 ± 131.06
**Phase 8**	**HRP**	38.58 ± 0.65	38.58 ± 0.72	38.23 ± 0.53	530.43 ± 65.70
	**SCENARIO**	-	37.80 ± 0.59	38.40 ± 0.60	471.15 ± 127.73
	**HuTheReg**	39.80 ± 0.69	39.44 ± 0.59	39.73 ± 0.66	590.36 ± 92.78
	**SCTM**	39.79 ± 1.35	39.51 ± 2.05	39.56 ± 1.28	404.50 ± 109.58

Note: HRP, health-risk prediction model; HuTheReg, human thermal regulation model; SCTM, six-cylinder thermoregulatory model; *T_Brain_*, brain temperature; *T_Skin_*, skin temperature; *T_Core_*, core body temperature; *ER*, sweat evaporation rate.

**Table 6 ijerph-19-07950-t006:** Correlation between models predictions for moderate working profile G_12.

		HRP	SCENARIO	HuTheReg	SCTM
**T_Brain_ (°C)**	**HRP**	1.00	-	0.86	0.91
	**SCENARIO**	-	-	-	-
	**HuTheReg**	0.86	-	1.00	**0.96**
	**SCTM**	0.91	-	**0.96**	1.00
**T_Skin_ (°C)**	**HRP**	1.00	0.81	0.92	**0.97**
	**SCENARIO**	0.81	1.00	0.89	0.89
	**HuTheReg**	0.92	0.89	1.00	0.95
	**SCTM**	**0.97**	0.89	0.95	1.00
**T_Core_ (°C)**	**HRP**	1.00	0.90	0.93	0.90
	**SCENARIO**	0.90	1.00	**0.96**	0.91
	**HuTheReg**	0.93	**0.96**	1.00	0.95
	**SCTM**	0.90	0.91	0.95	1.00
**ER (W)**	**HRP**	1.00	0.96	0.93	0.83
	**SCENARIO**	0.96	1.00	**0.98**	0.91
	**HuTheReg**	0.93	**0.98**	1.00	0.96
	**SCTM**	0.83	0.91	0.96	1.00

**Table 7 ijerph-19-07950-t007:** Correlation between models predictions for stressful working profile G_14.

		HRP	SCENARIO	HuTheReg	SCTM
**T_Brain_ (°C)**	**HRP**	1.00	-	0.87	0.90
	**SCENARIO**	-	-	-	-
	**HuTheReg**	0.87	-	1.00	**0.96**
	**SCTM**	0.90	-	**0.96**	1.00
**T_Skin_ (°C)**	**HRP**	1.00	0.96	0.88	**0.97**
	**SCENARIO**	0.96	1.00	0.95	**0.97**
	**HuTheReg**	0.88	0.95	1.00	0.94
	**SCTM**	**0.97**	**0.97**	0.94	1.00
**T_Core_ (°C)**	**HRP**	1.00	**0.98**	0.92	0.84
	**SCENARIO**	**0.98**	1.00	0.94	0.88
	**HuTheReg**	0.92	0.94	1.00	0.88
	**SCTM**	0.84	0.88	0.88	1.00
**ER (W)**	**HRP**	1.00	0.94	0.94	0.87
	**SCENARIO**	0.94	1.00	**0.98**	0.90
	**HuTheReg**	0.94	**0.98**	1.00	0.88
	**SCTM**	0.87	0.90	0.88	1.00

Note: Highlighted/bold cells indicate highest correlation for the variable of interest and condition.

## Data Availability

Data from this analysis have been obtained through sharing agreements and therefore must be coordinated for use by the originators.

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
