# Peer review of "Use of Thermoregulatory Models to Evaluate Heat Stress in Industrial Environments"

_ijerph, 2022, doi:10.3390/ijerph19137950_

Round 1

Reviewer 1 Report

The manuscript presented for review is primarily descriptive and contains a comparison of four developed thermoregulatory models for their use in the prevention and alleviation of the effects of heat stress.

The manuscript is at a good level when it comes to the originality of the solutions presented and the final conclusions formulated, and its results can be used to more fully predict heat stress during exposure to hot environments and will certainly meet with the interest of readers.

Although the manuscript contains only the analysis of four thermoregulatory models to better predict temperatures  of brain, skin, core and rate of sweat evaporation, it brings valuable insights into the discussion and points directions for further research related to ensuring the protection of a larger population of individuals from heat injuries and life-threatening.

The manuscript presented in this form is practically eligible for publication in the IJERPH, only in order to improve its readability it is recommended to introduce the List of Abbreviations.

Author Response

Thank you for your thoughtful review!  We have added an abbreviation list at the beginning of the manuscript. 

Additionally, we have uploaded the full set of reviewer comments and our responses to provide you visibility.

Reviewer 2 Report

I read the manuscript with great interest. The authors presented thermoregulatory models to evaluate heat stress in industrial environments. The manuscript includes detailed methodological results. It can be considered publication after considering the following major points:

1) The Abstract should be harmonized and needs serious revision. Those words indicating, the result, and conclusion should be removed, and present the main findings.
2) The introduction needs more up-to-date information. Meanwhile, the gaps that the present study will expect to fill should also be included. 
3) In Table 4, moderate working profile G_12, Phase 2, the deviation of Health Risk Prediction is higher than the normal evaluation. why? Please provide a valuable description. 
4) The whole manuscript needs critical revisions, such as English, unnecessary space, abbreviation, etc.
5) I suggest the authors to consider up-to-date references throughout the manuscript.

Author Response

Thank you for your thoughtful review.  Below are the specific responses. Additionally, we have uploaded the full set of reviewer comments and our responses to provide you visibility.

Reviewer 2

I read the manuscript with great interest. The authors presented thermoregulatory models to evaluate heat stress in industrial environments. The manuscript includes detailed methodological results. It can be considered publication after considering the following major points:

1) The Abstract should be harmonized and needs serious revision. Those words indicating, the result, and conclusion should be removed, and present the main findings.
Author Response: Thank you!  This is a great suggestion.  We have removed the heading words and have revised the abstract.

2) The introduction needs more up-to-date information. Meanwhile, the gaps that the present study will expect to fill should also be included. 
Author Response: This is a good point!  We have added to the introduction and included some newer information and improved upon how we conveyed the research gaps addressed in the work.  

3) In Table 4, moderate working profile G_12, Phase 2, the deviation of Health Risk Prediction is higher than the normal evaluation. why? Please provide a valuable description. 
Author Response: Thank you; this is an interesting question.  The deviations of the HRP model temperatures are actually lower than the other models; while the evaporation rate (ER) is higher.  The increased ER in the model seems to stabilize the temperature predictions more compared to the other models.  Until we run some new comparisons between the models and some actual data, we will be unable to tell if this is helpful for accuracy or not.  We have added a note regarding this point to the discussion.

4) The whole manuscript needs critical revisions, such as English, unnecessary space, abbreviation, etc.
Author Response: We have carefully gone through the manuscript and improved clarity and readability.  We have included an abbreviation list at the beginning of the manuscript to help readability, have reduced any unneeded spaces, and have made grammatical revisions throughout.

5) I suggest the authors to consider up-to-date references throughout the manuscript.
Author Response: Thank you.  We have expanded upon the introduction and discussion to include a wider range of references.

Reviewer 3 Report

The paper deals with the evaluation of different thermal stress models.

The topic is interesting however tha paer should be improved to be considered suitable for the publication.

The aims of the paper should be explained at the end of the introduction, while in the current version there is more than anything else a summary of the activities carried out.

Section 2

It is not clear how the working profiles G12 and G14 differ, plese better explain the two profile.

In Table 2, the left and right parts are identical except for the last row. In Table 3 the rows are all equal in pairs. It is suggested that these tables be reconsidered or replaced with a description in the text.

In the units please do not use. as a symbol of multiplication.

As the aim of the work is not clear, discussions and conclusions should also be reconsidered.

The conclusions should be improved and provide guidance based on the results obtained, while in the current version they are too hurried.

If the objective of the work is to compare the evaluation models of heat stress in the conclusions, considerations in this area and any consequent implications are expected.

Reference:

The list of references seems very limited to the authors; being a very studied topic, I believe that authors should broaden their citation landscape to include works by other research groups who have worked on the topic.

Author Response

Thank you for your thoughtful review.  Below are the specific responses. Additionally, we have uploaded the full set of reviewer comments and our responses to provide you visibility.

Reviewer 3

The paper deals with the evaluation of different thermal stress models.

The topic is interesting however tha paer should be improved to be considered suitable for the publication.

The aims of the paper should be explained at the end of the introduction, while in the current version there is more than anything else a summary of the activities carried out.

Author Response: Thank you, this is a good point. Reviewer 2 also mentioned this. We have revised the introduction and included some newer information and references; we have also improved upon how we conveyed the research gaps addressed in the work.

Section 2

It is not clear how the working profiles G12 and G14 differ, plese better explain the two profile.

In Table 2, the left and right parts are identical except for the last row. In Table 3 the rows are all equal in pairs. It is suggested that these tables be reconsidered or replaced with a description in the text.
Author Response: Thank you; this is a helpful suggestion.  We have revised both of these tables to be more straightforward and removed any redundancies.

In the units please do not use. as a symbol of multiplication.
Author Response: We have corrected these to be the appropriate multiplication symbol (·).

As the aim of the work is not clear, discussions and conclusions should also be reconsidered.
Author Response: Thank you; we have revised the text throughout to clarify the goals of the work and specifically within the introduction, discussion, and conclusions.

The conclusions should be improved and provide guidance based on the results obtained, while in the current version they are too hurried.

If the objective of the work is to compare the evaluation models of heat stress in the conclusions, considerations in this area and any consequent implications are expected.
Author Response: This is a good point!  We have revised the conclusion to reflect the answers and observations from the analyses.

Reference:

The list of references seems very limited to the authors; being a very studied topic, I believe that authors should broaden their citation landscape to include works by other research groups who have worked on the topic.
Author Response: Thank you; this is a very good point!  We have included a wider range of references to broaden and highlight the valuable work of others working in this area.

Round 2

Reviewer 2 Report

The authors revised the manuscript in line with the comments. It can be considered for publication.

Author Response

Thank you for your thoughtful review!

Reviewer 3 Report

The authors revised the manuscript according to all my suggestions. In my opinion, the paper can now be considered for publication.

Author Response

Thank you for your thoughtful review!